# Design and Characterization of a Novel Biaxial Bionic Hair Flow Sensor Based on Resonant Sensing

**DOI:** 10.3390/s20164483

**Published:** 2020-08-11

**Authors:** Zhuoyue Liang, Xin Guo, Bo Yang, Ting Zhang

**Affiliations:** 1School of Instrument Science and Engineering, Southeast University, Nanjing 210096, China; 220183262@seu.edu.cn (Z.L.); 230179247@seu.edu.cn (X.G.); 220183214@seu.edu.cn (T.Z.); 2Key Laboratory of Micro-Inertial Instrument and Advanced Navigation Technology, Ministry of Education, Nanjing 210096, China

**Keywords:** hair flow sensor, digital phase locked-loop, oscillatory flow velocity, amplitude closed-loop

## Abstract

This paper presents the design, theoretical analysis, simulation verification, fabrication and prototype characterization of a novel biaxial bionic hair flow sensor based on resonant sensing. Firstly, the device architecture, mainly consists of a polymer hair post, a silicon micro signal transducer and a glass substrate, is described, the theoretical simplified model is established and the mechanical sensitivity to air flow is deducted. Then, the structure simulations based on Ansys software are implemented to preliminarily verify the feasibility of the proposed sensor conception and optimize the structure parameters simultaneously. Subsequently, a closed-loop control scheme based on digital phase-locked loop and an amplitude demodulation algorithm of oscillatory flow velocity based on the least mean square method are proposed to transform and extract the air flow signal, and then verify it by circuit simulations based on SIMULINK. Finally, the fabricated prototype is illustrated and comprehensively tested. The tested prototype possesses an x-axis scale factor of 1.56 Hz/(m/s)^2^ and a y-axis scale factor of 1.81 Hz/(m/s)^2^ for the steady air flow and an x-axis detection threshold of 43.27 mm/s and a y-axis detection threshold of 41.85 mm/s for the oscillatory air flow.

## 1. Introduction

Cilia are hair-like microstructures found in insects and other living organisms. Biologists claim that creatures evaluate their extremely sensitive cilia system to extract and transmit physical signals to biological mechanoreceptors, so as to measure the external water or air flow velocity, acceleration and even angular velocity, and finally to assist the motion attitude control of natural organisms [1]. Subsequently, researchers and engineers believed that bionic hair sensors are a promising solution to realize flow measurement with high sensitivity, large dynamic range and low power consumption [2,3]. Various approaches, such as the capacitive [4,5], piezoelectricity, piezoresistance [6,7,8,9,10] and photoelectricity, are also be utilized to detect the ciliary bending moment derived from external air flow. In [11], a novel flow sensor based on resonant sensing with a two-stage microleverge mechanism was designed. The rotating beam prevents the eccentric rotation of the main frame when the resistance is transferred to the main frame, which constitutes the hinge of the first stage leverage. Through the traditional second lever, it forms the two-stage microleverage mechanism, which aims to increase the sensitivity of the signal detector by amplifying the resistance. In [12], inspired by the fish lateral line system, an artificial lateral line system based on the bionic hair sensor with a resonance readout was designed by using this structure. Compared with the signal from a single hair releasing sensor, the biaxial velocity measurement is realized by different array distributions. The test results show that the parallel array of four sensors can realize the measurement of constant air flow velocity and oscillation air flow in a strong DC environment. In [13], M. Dijkstra et al. developed a capacitive hair flow sensor with a sensitivity of 0.2 pF/V. The hair posts are installed on a sensitive capacitive membrane, and the drag force derived from external flow results in membrane deflection, thus the external flow detection can be realized by measuring the drift of membrane capacitance. A piezoresistive hydrophone inspired by the lateral line of fish was designed in [14,15,16]. The resistance change of the varistor caused by sound pressure in deep water can be detected by artificial hair. The tested hydrophone illustrates a sensitivity of −200 dB in the frequency below 800 Hz, and a regular figure of eight pattern is shown in the direction experiment. Krijnen et al. introduced a biomimetic flow sensor for environmental awareness in [17]. The input acceleration is detected by measuring the differential capacitance shift. The experiment results show that the fabricated prototype has a sense mode frequency of 367 Hz, a detection threshold of 0.63 m/s^2^ and a maximum dynamic range of 1 g. In [18], a photoelectric hair sensor consisting of artificial cilia supported by an elastic membrane was demonstrated. Light-emitting diodes and quadrant photodiodes are utilized as photoelectric sensitive components. Air flow or pressure waves directly stimulate cilia sway which is then detected by quadrant photodiodes. The superior resolution is illustrated in tested prototypes due to high sensitivity of photoelectric sensors. For the water flow measurement, a biological lateral line sensor consisting of an in-plane fixed cantilever and vertical cilia was reported in [19]. The cantilever and cilia are rigidly connected and perpendicular to each other, where the vertical cilia convert the magnitude of the water flow velocity into the strain shift of the cantilever beam. A distributed array of presented sensors is proposed to implement the flow field mapping and the tested prototype demonstrates a measurable water flow velocity range of 0.1~1 m/s and a resolution of 0.5 mm/s.

This work presents a novel biaxial bionic hair flow sensor based on resonant sensing. The device description is presented in Section 2, followed by the simulation analysis in Section 3. Then the experimental results of fabricated prototypes and the corresponding discussion are described in Section 4. Finally, Section 5 concludes this paper with a summary of the results.

## 2. Device Description

The structure schematic of the novel biaxial bionic hair flow sensor based on resonant sensing is illustrated in Figure 1. The proposed device is mainly composed of three parts: a polymer hair post, a silicon micro signal transducer and a glass substrate. The interaction with the environment is achieved via the hair post. To realize the dual-axis sensitivity of air flow, the silicon micro signal transducer consists of a main frame, four elastic structures and four sub-sensitive structures, which are respectively arranged on the in-plane axes of the silicon micro signal transducer. Four identical square-wave elastic structures are designed for the suppression of excessive structural out-of-plane swing and the deformation of proposed elastic beams produces elastic forces opposite to the main frame movement direction, which can effectively suppress the structural swing in the z-axis direction and improve the stability of the main frame. Four identical resonators are arranged in four in-plane directions for direct bi-directional input air flow sensing. The adopted sub-sensitive structures consist of a micro lever, a double-ended tuning fork (DETF), driving combs and detection combs. The micro lever is utilized to amplify the deflection force derived from the external fluid, and the leverage structure parameters are optimized based on the finite element method (FEM) simulation to obtain a maximum force magnification. The structure parameters are shown in Table 1. The sensor we propose in this paper, based on the quasi-digital resonant sensing, is able to detect the in-plane biaxial air flow with a single silicon signal transducer, which is beneficial for the improvement of structural space utilization. The silicon main frame is dragged to deflect by a certain angle when the polymer hair post bears the drag force from the external flow field, and the natural frequency of the resonator shifts simultaneously. The vibration amplitude of the resonator beams reaches the maximum when the frequency of the electrostatic drive force matches the natural frequency of the resonator beams, and the phase of electrostatic drive force lags the vibration displacement by 90 degrees. Therefore, a closed-loop self-excitation circuit is designed to lock the natural frequency by controlling the phase of the applied driving force. Meanwhile, a digital proportional integral (PI) control circuit is designed to construct an amplitude closed-loop system to stabilize the vibration amplitude. Finally, the input of air flow can be measured by detecting the frequency change of the closed-loop self-excitation system [20,21].

### Air Flow Measurement

Suppose the drag force exerted on the hair post is [22]
(1)$$F_{h}=\frac{1}{2}C_{D}\rho u^{2}A=\frac{1}{2}C_{D}\rho u^{2}DL_{h}$$
where *F_h_* represents the drag force of hair in the flow field, *C_D_* denotes the drag coefficient, *ρ* denotes the air density, *u* represents the velocity of air, *A* indicates the area of hair in the direction of air inflow, *D* is the diameter of the hair post and *L_h_* is the length of the hair post.

The force acting on the resonant beams of the DETF is estimated as
(2)$$F=\eta A_{1}F_{h}$$
where *F_h_* represents the drag force of hair in the flow field, *A*_1_ denotes the amplification coefficient and *η* denotes the attenuation coefficient. According to the vibration theory [23,24], the natural frequency of the tuning fork resonant beam is
(3)$$\omega_{n}=\sqrt{\frac{K_{eff}}{M_{eff}}}$$
where *K_eff_* represents the equivalent stiffness of the tuning fork and *M_eff_* represents the equivalent mass of the tuning fork. When the resonant beam is subjected to an axial force, the stiffness variation is proportional to the axial force. Then the natural frequency can be expressed as
(4)$$\omega_{n}^{\prime }=\sqrt{\frac{K_{eff}+\delta F}{M_{eff}}}=\sqrt{\frac{K_{eff}}{M_{eff}}+\frac{\delta }{M_{eff}}F}$$
where *δ* is a constant associated with structural parameters. By Taylor series expansion, Equation (4) can be rewritten as
(5)$$\omega_{n}^{\prime }=\omega_{n}(1+\frac{1}{2}tF-\frac{1}{8}t^{2}F^{2}+\frac{1}{16}t^{3}F^{3}-\ldots )$$
where *t* is a constant associated with the tuning fork. According to Formulas (1) and (2), the frequency variation of the resonator is obtained by ignoring the influence of high order nonlinearity, that is
(6)$$\Delta \omega_{n}=\omega_{n}^{\prime }-\omega_{n}=\pm \frac{1}{2}\omega_{n}tF=\pm \frac{1}{4}\omega_{n}t\eta A_{1}C_{D}\rho u^{2}DL_{h}$$

Two resonators in the same axial direction are forced in the opposite direction, which results in the opposite change of natural frequency. As illustrated, the resonator frequency difference Δ*ω_n_* is proportional to the applied axial force, which means the frequency difference of two resonators is proportional to the square of the input air flow velocity [25,26,27]. The input–output relationship between the air flow velocity and frequency difference, namely the sensitivity of the proposed sensor, can be expressed as Formula (7).
(7)$$S_{\omega }=\frac{d\omega }{du^{2}}=\frac{(\omega_{01}-\;\omega_{02})C_{D}\rho DL_{h}\delta }{4}=\frac{\omega_{0}C_{D}\rho DL_{h}\delta }{2}$$
(8)$$\delta =tA_{1}\eta$$
where *ρ* denotes the air density, *u* represents the velocity of air, *D* is the diameter of the hair post, *L_h_* is the length of the hair post, *ω*_01_ and *ω*_02_ represent the angular frequencies of two symmetric resonators, *δ* represents the constant associated with the device structure, *A*_1_ denotes the amplification coefficient, *η* denotes the attenuation coefficient and *t* is a constant associated with the tuning fork. The theoretical structural parameters are listed in Table 1. As illustrated, the proposed device has a theoretical mechanical sensitivity of 1.583 Hz/(m/s)^2^. The effective electrostatic driving force *F_d_* provided to the drive combs is estimated as
(9)$$F_{d}=\frac{nh\varepsilon }{d}\cdot V^{2}$$
where *n* is the number of combs, *ε* is the dielectric constant of air, *h* is the height of combs and *d* is the distance of combs. Thus, the driving voltage is expressed as
(10)$$V=V_{d}+V_{a}sin\left(\omega_{d}t\right)$$
where *V_d_* represents the direct current bias voltage and *V_a_* and *ω_d_* represent the amplitude and frequency of the applied alternating voltage, respectively. Substituting Equation (10) into Equation (9), the effective electrostatic driving force *F_d_* can be rewritten as
(11)$$F_{d}=\frac{nh\varepsilon }{d}\cdot \left(V_{d}^{2}+2V_{a}V_{d}sin\left(\omega_{d}t\right)+\frac{1}{2}V_{a}^{2}-\frac{1}{2}V_{a}^{2}cos\left(2\omega_{d}t\right)\right)$$

When *V_d_* >> *V_a_*, the influence of high-order items is ignored and Equation (11) is simplified as
(12)$$F_{d}=\frac{nh\varepsilon }{d}\cdot 2V_{a}V_{d}sin\left(\omega_{d}t\right)$$

According to Equation (12), the amplitude of the effective driving force *F_d_* is proportional to *V_d_*·*V_a_*.

## 3. Simulation and Optimization

Finite element modeling (FEM) simulation is implemented to evaluate and optimize the performance of the proposed structure design in this section.

### 3.1. Mode Simulation

The mode simulation is utilized to evaluate the mode shapes and natural frequencies of the proposed structure. The effective working modes of the proposed sensor are shown in Figure 2 and the natural frequencies of effective modes are listed in Table 2. As illustrated in Figure 2A,B, the first and second modes of the device are designed as the sense modes, and the mode frequencies are 344.19 Hz and 344.49 Hz, respectively. It is obvious that the sense mode frequencies are relatively low, which is conducive to improving the mechanical sensitivity. The frequency difference of the first and second modes can be attributed to the asymmetry of structure meshing and simulation calculation error. Similarly, there is a certain error in the simulation of the frequencies of resonators. The interference mode means the invalid mode. The interference mode is not stimulated when the device is working. Due to the better damping characteristics of DETFs, the anti-phase mode is selected as the operation mode of DETFs. The resonant beams of DETFs have identical vibration amplitudes and the opposite movement direction in the anti-phase mode. The natural frequencies of the four resonators in the anti-phase mode are 25,106 Hz, 25,108 Hz, 25,109 Hz and 25,113 Hz respectively, as shown in Figure 2C–F. Additionally, the natural frequencies of the interference modes are 23,182 Hz and 28,789 Hz, respectively, as show in Figure 2G,H. The minimum frequency difference between the resonant operating mode and the interference mode is 1924 Hz and 3676 Hz, therefore, the simulation results confirm that the effective modes and the interference modes are effectively isolated.

The key conversion mechanism of the proposed hair flow sensor is the polymer hair post. The increase in the height or diameter increases the force-bearing surface and further improves the device sensitivity. Therefore, the variations of resonant frequency with different hair dimensions are simulated, as illustrated in Figure 3. When the hair height increases from 1000 µm to 10,000 µm (the diameter is 1000 µm), the frequency variation of the DETF working mode increases 12.77 times, 13.7 times and 8.45 times with an input air flow of 0.1 m/s, 1 m/s and 10 m/s, respectively. Additionally, after increasing the hair diameter from 100 µm to 1000 µm (the height is 9000 µm), the frequency variation of the DETF working mode increases 7.3 times, 5.36 times and 5.2 times, respectively. The simulated results reveal that both the increase in hair height and diameter improve the sensitivity of the sensor, which is consistent with the theoretical analysis in Formula (6). Therefore, the mechanical sensitivity of the proposed flow sensor can be optimized by adjusting the hair dimensions.

The relationship between output frequency and flow velocity at three different hair heights are shown in Figure 4. For the x-axis input air flow, the corresponding sensitivities are 0.59 Hz/(m/s)^2^, 1.16 Hz/(m/s)^2^ and 1.69 Hz/(m/s)^2^ with a hair height of 3000 µm, 6000 µm and 9000 µm. Similarly, the corresponding sensitivities are 0.53 Hz/(m/s)^2^, 1.07 Hz/(m/s)^2^ and 1.64 Hz/(m/s)^2^ for the y-axis input air flow. The results also confirm that the output frequency difference is positively correlated with the square of the air flow velocity, which is also consistent with the theoretical analysis results.

The cross-axis coupling derived from the orthogonal error is the main obstacle that limits the accuracy of the proposed biaxial sensor. As shown in Figure 5, when the flow velocity acts in the x-axis direction, the simulated x-axis maximum sensitivity to air flow with a 9000 µm hair height is 1.69 Hz/(m/s)^2^ and the corresponding y-axis sensitivity is 0.058 Hz/(m/s)^2^, which indicates a coupling coefficient of 0.033 in the x-axis. Similarly, a coupling coefficient of 0.025 is simulated in the y-axis. Therefore, the influence of cross-axis coupling can be ignored.

### 3.2. Circuit Simulation

The closed-loop control circuits of phase and amplitude are designed to extract the natural frequency of DETFs accurately and effectively. Firstly, the excitation signals are applied on the drive electrodes and the extracting detection signals are extracted from the sense electrodes [28,29]. Then a C/V converter is adopted to transform the extracted capacitance signal into a voltage signal. The phase of detected vibration displacement is synchronized with that of the driving signal by the phase-locked loop (PLL), consisting of a phase frequency detector (PFD), a low pass filter (LPF) and a digitally controlled oscillator (DCO) [30]. Accordingly, the phase-locked loop in the field programmable gate array (FPGA) chip can also be utilized as a narrow-band filter to further improve the measurement accuracy of the hair sensor. The following design of the amplitude loop is critical for the measurement performance. In the amplitude control loop, the detection signal of the detection combs is rectified into a full wave and transferred to the filter circuit. The LPF is utilized to filter out the higher order frequency and the direct current value *V_a_*, representing the amplitude of the detection signal, is obtained, which can be kept stable by a PI controller. The alternating current component of the driving signal is obtained by multiplying the signal behind the controller with the output signal of the phase control loop, and then the alternating current signal is added to *V_d_* to obtain the final driving signal. The PI controller is utilized to reduce the adjustment time and error while eliminating the steady-state error of the system and stabilizing the vibration amplitude at the reference voltage. Subsequently, the frequency of the excitation signals is locked to the natural frequency of the DETFs and the amplitude remains stable at a custom value. The block diagram of the control and measurement system is shown in Figure 6.

The control circuit simulation is implemented by SIMULINK software. The adopted parameters are listed in Table 3, and the oscillograms of key signals, namely the output of the interface amplifier, the applied driving voltage of the sensor, the output signal of the loop filter and the output signal of the amplitude loop, are plotted in Figure 7. Experimental results indicate that the vibration amplitude reaches the maximum when the driving signal frequency is close to the natural frequency of the DETF. From Figure 7, it is obvious that, after a control period of about 0.25 s, the output of the interface amplifier is finally stabilized at about 2.3 V and the frequency of the driving voltage is finally locked at the natural frequency of the DETFs. The locked signal after the PI controller of the amplitude control loop is utilized to control the driving signal. The output signal of the PI controller reaches a stable value of 1.7 V, and the amplitude of the driving voltage is finally stabilized at the reference value of 3.5 V. The effectiveness of the control system is further verified.

### 3.3. Demodulation Algorithm

When the proposed hair sensor is arranged in the oscillatory air flow, the frequency difference of the DETFs is also an alternating current signal that requires a specific demodulation scheme [12]. In this paper, a demodulation approach based on the minimization criterion of the mean square error is proposed to extract the effective signal from the noise signal, as shown in Figure 8. The criterion parameters of the mean square error can be automatically adjusted according to the characteristics of the input signal to meet the requirements of the optimal filtering criterion. The error algorithm of the minimum mean square greatly simplifies the complexity of the calculation since the auto-correlation matrix inversion calculation is not necessary. Therefore, the error algorithm of the minimum mean square is utilized to demodulate the oscillatory air flow. After filtering the direct current in the signal, the input signal *a*(*n*) of the oscillatory air flow with noise is obtained. The predicted vector *p*(*n*) is multiplied by the reference signal *c*(*n*) to obtain the predicted signal *b*(*n*), and *c*(*n*) is an orthogonal digital reference signal, which is directly generated by the coordinate rotation digital computer (CORDIC) algorithm. The variable *e*(*n*) is the difference between the input signal *a*(*n*) and the predicted signal *b*(*n*), and its mean square value is minimized through continuous iteration in the criterion, which makes *a*(*n*) closer to *b*(*n*) [31]. Hence, the input oscillatory signal is finally demodulated.

An orthogonal digital reference signal *c*(*n*) is generated by FPGA.
(13)$$c(n)=\left[\begin{array}{l} sin\omega_{c}t\\ cos\omega_{c}t \end{array}\right]$$
where *ω_c_* is the frequency of the oscillatory flow. 

The adopted iteration scheme is mainly based on the steepest descent method in the optimal algorithm, and the gradient estimation is utilized to realize a fast search instead of a gradient vector. The following formula represents the recursive formula of the steepest descent algorithm and the next prediction matrix *p*(*n +* 1) is obtained by adding the previous prediction matrix *p*(*n*) and the scale modified negative gradient estimation.
(14)$$p(n+1)=p(n)-\mu \nabla E[e^{2}(n)]\approx p(n)+2\mu e(n)c(n)$$
where *μ* is the iteration step and *E*[*e*²(*k*)] represents the mean of the squared error signal, which is replaced by the instantaneous square value of the error signal in each iteration. After demodulation, an optimal matrix *p* is obtained by an iterative algorithm, which can be expanded to
(15)$$p={\left[\begin{array}{cc} Q & Ι \end{array}\right]}^{Τ}$$
where *Q* and *I* are the values obtained by the iterative algorithm.
(16)Q=2Acosθ
(17)I=2Asinθ
where *A* represents the amplitude of the demodulated signal. According to Formulas (16) and (17), the amplitude of the input signal can be expressed as
(18)$$A=\sqrt{\frac{Q^{2}+I^{2}}{4}}$$

The simulations of the demodulation algorithm are presented for various input signals, as demonstrated in Figure 9. In the MATLAB simulation experiment, the input signal is superposed by a 10 V Gaussian white noise and a 2 V oscillatory flow output signal of the resonator.

As illustrated in the results, the convergence speed and accuracy of the algorithm are affected by the iteration step, as with the decrease in the iteration step, the convergence accuracy increases and the convergence rate decreases. Therefore, on the premise of ensuring the accuracy, the optimal iteration step should maintain a relatively short adaptive time. It is worth mentioning that the signal amplitude can still be demodulated when the amplitude of the noise signal is five times the useful signal, which means that the least mean square demodulation algorithm can be utilized to extract the amplitude of the oscillatory flow signal.

## 4. Results and Discussion

The optical photographs of fabricated prototype are demonstrated in Figure 10. The fabricated silicon transducer has a dimension of 10,000 µm (length) × 10,000 µm (width) × 100 µm (height), and the traditional deep dry silicon on glass (DDSOG) micro-electro-mechanical system (MEMS) process is adopted to fabricate the proposed hair flow sensor. Three photoresist masks are utilized to pattern the signal wires, anchors and sensitive structures. A monocrystalline wafer with a thickness of 200 microns and a Pyrex glass wafer with a thickness of 500 microns are processed to fabricated the designed sensitive structure. First, the signal wires are sputtered after a wet etching process with buffered oxide etch (BOE) solution on the Pyrex glass substrate based on the first photoresist mask. Then, the anchors with a step height of 20 microns are etching on a polished silicon wafer via a deep silicon dry etching process and then attached to the glass substrate based on the standard anodic bonding process. After that, the thickness of the monocrystalline wafer is decreased to 100 microns by a wet etching process with potassium hydroxide solution. Subsequently, the monocrystalline wafer is lithographically patterned based on the third photoresist mask and etched to release the mechanical structures via the deep reactive ion etching (DRIE). Finally, a polymer hair post is micro-assembled on the silicon transducer using UV glue.

The peripheral hardware, consisting of an interface board, an A/D-D/A transformation board and an FPGA board, is fabricated, as shown in Figure 11. The weak air flow signal derived from the hair sensor is extracted and transformed by the interface board, then converted to digital signals via the A/D-D/A transformation board, and finally processed by the FPGA board.

### 4.1. Steady Air Flow Detection Experiment

The measurement setup for steady air flow performance evaluation is shown in Figure 12. The FPGA chip is programmed by the JTAG/AS interface. A fan is adopted to create the steady air flow which is calibrated by a commercial hot-film anemometer. The least squares method is used to fit the relationship between the frequency difference of the resonator and the square of the input air flow in two sensitive axes.

Since the frequency shift is directly proportional to the square of the air velocity, the square of the velocity ((m/s)^2^) is adopted as the abscissa. The test results of the fabricated prototypes are demonstrated in Figure 13. The black dots represent the measured output signals and the red lines are the fitted curves based on the least squares method. As illustrated in the experiment results, the fabricated hair flow sensor has an x-axis scale factor of 1.56 Hz/(m/s)^2^ and a y-axis scale factor of 1.81 Hz/(m/s)^2^, which shows the sensitivity of the hair sensor to the air flow velocity. The scale factor of the x-axis is close to the theoretical calculation (sensitivity difference of 0.02 Hz/(m/s)²), while the deviation between the tested and calculated scale factors of the y-axis is relatively large (sensitivity difference of 0.22 Hz/(m/s)²). The sensitivity difference in the two directions can be mainly attributed to the install error of the hair post, which lead to the error of transmitted axial force. Moreover, the tested natural frequencies of resonant beams 23,706 Hz, 23,813 Hz, 24,225 Hz and 23,396 Hz are not exactly the same with the theoretical designed frequency 25,039 Hz and simulated frequency 25,106 Hz, 25,108 Hz, 25,109 Hz and 25,113 Hz, which indicates the existence of fabrication error of silicon structure and also lead to the mechanical sensitivity difference. In addition, the inadequate accuracy of measurement and control circuit and the uncertainty of generated air flow also affect the final tested mechanical sensitivity.

The nonlinearity is characterized by the ratio of the maximum deviation of the system output velocity to the measured velocity range. The hair flow sensor has an x-axis nonlinearity of 9.175% and a y-axis nonlinearity of 7.626%, which shows a good coincidence between the measured data and the fitting data.

### 4.2. Oscillatory Air Flow Detection Experiment

The measurement setup for oscillatory flow performance evaluation is shown in Figure 14. The particle velocity refers to the reciprocating oscillation of air particles under the stimulation of oscillatory air flow. Based on the wave equation, the particle vibration velocity can be obtained indirectly by measuring the sound pressure [32,33]. A commercial speaker is utilized to generate the oscillatory air flow with a frequency of 60 Hz, and increase the particle vibration velocity from 0 to 80 mm/s. The oscillatory flow velocity is calibrated by a microphone and the relationship between the sound pressure of plane wave *P_e_* and the particle vibration velocity *V_x_* can be expressed as
(19)$$V_{x}=\frac{P_{e}}{\rho_{0}c_{0}}\sin(\omega t-\frac{\omega }{c_{0}}x)$$
where *ρ*_0_ represents air medium density, *c*_0_ is the acoustic velocity in air, *ω* is the oscillation frequency and *x* is the distance from the particle to the sound source. As shown in Equation (19), the measured sound magnitude is proportional to the oscillatory air flow velocity. 

The measured oscillatory air flow after demodulation is shown in Figure 15. An optimum detecting threshold of 43.27 mm/s in the x-axis and 41.85 mm/s in the y-axis can be estimated by means of the least mean square demodulation algorithm. The experiment shows that the measured oscillatory air flow can be demodulated by the algorithm, and the frequency difference is positively correlated with the flow velocity over the threshold. Different from steady air flow detection, the oscillatory air flow measurement experiments aim to determine the detection thresholds. In the case of extremely low air flow velocity due to the boundary layer effect [11], the square linear relationship between the flow velocity and frequency shift is no longer valid, therefore, linear fitting cannot be performed. The output signal will be submerged in the noise and the useful signal cannot be extracted when the signal is less than the threshold velocity. The main performance indicators are summarized in Table 4.

## 5. Conclusions

The design, theoretical analysis, simulation verification, fabrication and prototype characterization of a novel biaxial bionic hair flow sensor based on resonator sensing is presented in this paper. Firstly, the device architecture, mainly consisting of a polymer hair post, a silicon micro signal transducer and a glass substrate, is described, the theoretical simplified model is established and the mechanical sensitivity to air flow is deducted. Then, the structure simulations based on ANSYS are implemented to preliminarily verify the feasibility of the proposed sensor conception and optimize the structure parameters simultaneously. Subsequently, a closed-loop control scheme, based on digital phase-locked loop and an amplitude demodulation algorithm of oscillatory flow velocity based on the least mean square method, are proposed to transform and extract the air flow signal, and then verify it by circuit simulations based on SIMULINK. Finally, the fabricated prototype is illustrated and comprehensively tested. The tested prototype possesses an x-axis scale factor of 1.56 Hz/(m/s)^2^ and a y-axis scale factor of 1.81 Hz/(m/s)^2^ for the steady air flow and an x-axis detection threshold of 43.27 mm/s and a y-axis detection threshold of 41.85 mm/s for the oscillatory air flow. The experimental results show that the proposed hair sensor scheme is promising in the field of micro velocity measurement. However, compared with the detection threshold of the hair flow sensor proposed by Prof. Gijs J. Krijnen in 2014 (352 um/s), the performance of the sensor proposed by our group is not comparable, not to mention the detection threshold of crickets in nature (73.2 um/s), which can be attributed to the insufficient precision of circuit processing and structural processing errors, etc. Compared with the same type of resonant hair flow sensor proposed by our group before [11] (mechanical sensitivity of 7.41 Hz/(m/s)^2^, detection threshold of 3.23 mm/s), the hair flow sensor proposed in this paper shows poorer evaluated performance since the effective sensitive structure space of the biaxial sensing architecture is less than that of the uniaxial device. Moreover, the cross-axis coupling between the two axes also reduces the signal-to-noise ratio of the measurement and control circuit, thereby affecting the sensitivity and measurement threshold. Despite all this, the proposed structure architecture realizes the measurement of in-plane two-axis air flow, which is beneficial for improving the space utilization of integrated air flow-sensitive equipment. Meanwhile, it provides a new potential solution for in-plane orientation estimation and positioning based on a single sensitive component. Our future work will focus on performance improvement and specific application research.

## Figures and Tables

**Figure 1 sensors-20-04483-f001:**
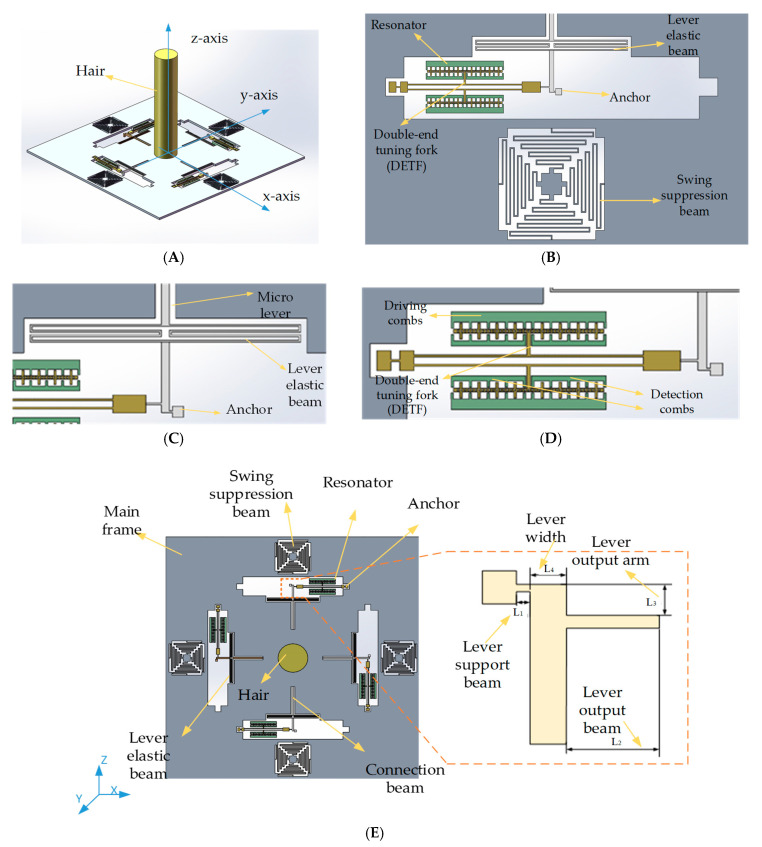
Schematic of the bionic hair flow. (**A**) The whole structure. (**B**) The double-ended tuning fork. (**C**) The lever elastic beam. (**D**) The resonator. (**E**) The silicon micro signal transducer.

**Figure 2 sensors-20-04483-f002:**
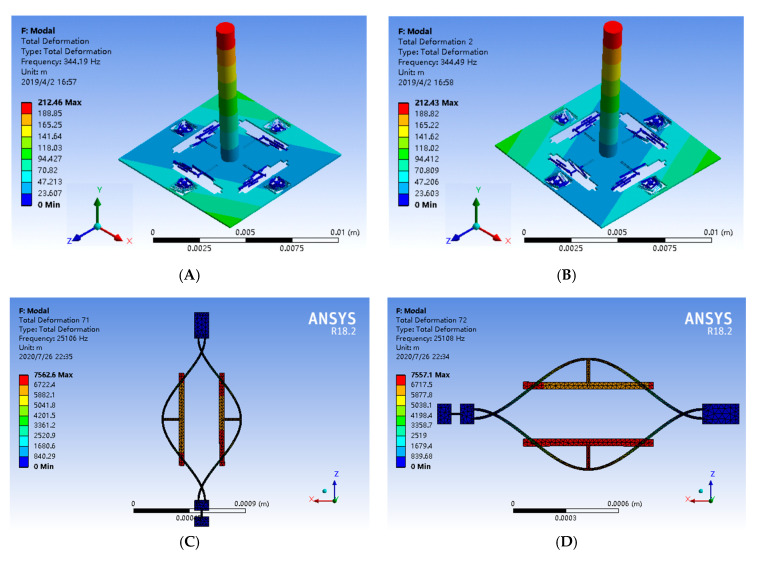
The mode simulation of the novel biaxial bionic hair flow sensor. (**A**) Sense mode 1. (**B**) Sense mode 2. (**C**) Working mode of left resonator. (**D**) Working mode of upper resonator. (**E**) Working mode of right resonator. (**F**) Working mode of lower resonator. (**G**) Interference mode 1. (**H**) Interference mode 2.

**Figure 3 sensors-20-04483-f003:**
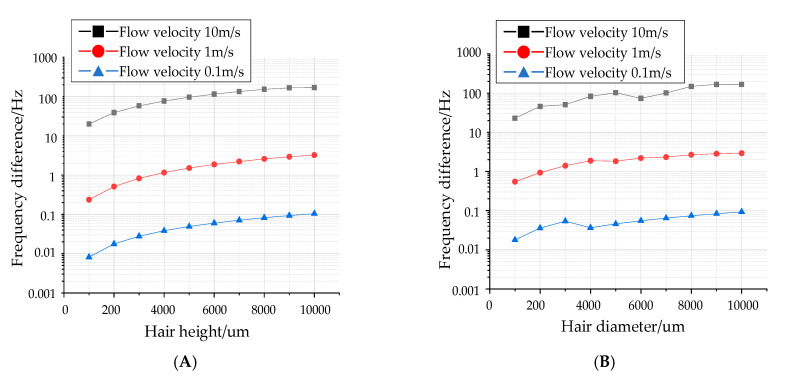
Influence of hair parameters on input and output characteristics. (**A**) Relationship between hair height and output frequency at different flow velocities. (**B**) Relationship between hair diameter and output frequency at different velocities.

**Figure 4 sensors-20-04483-f004:**
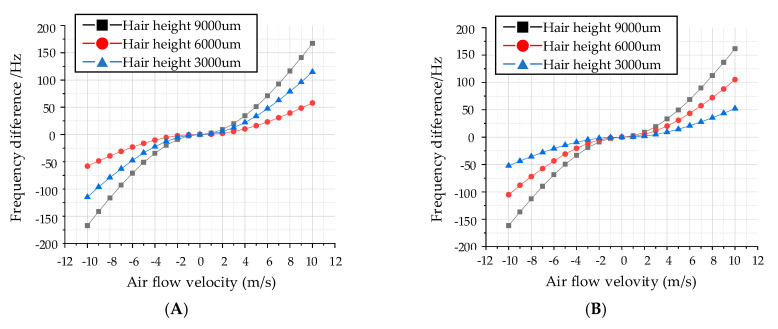
Relationship between output frequency and air flow velocity. (**A**) Output frequency difference in x-axis. (**B**) Output frequency difference in y-axis.

**Figure 5 sensors-20-04483-f005:**
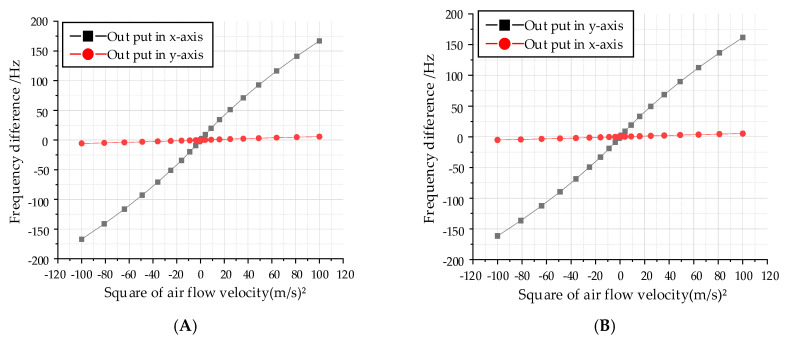
Influence of cross coupling on input and output characteristics. (**A**) The cross-coupling input air flow in the x-axis. (**B**) The cross-coupling input air flow in the y-axis.

**Figure 6 sensors-20-04483-f006:**
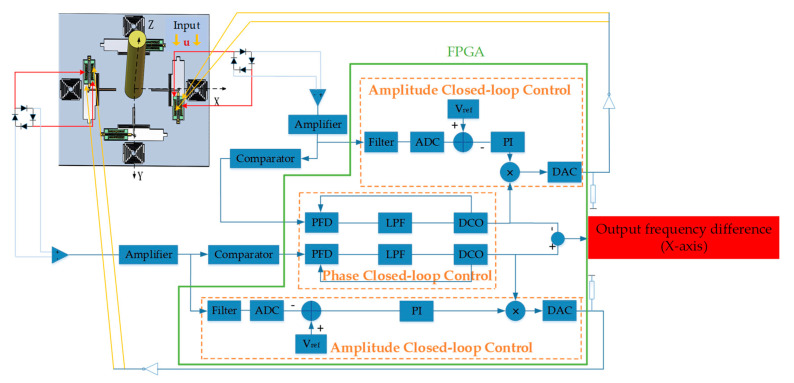
Diagram of the control and measurement system.

**Figure 7 sensors-20-04483-f007:**
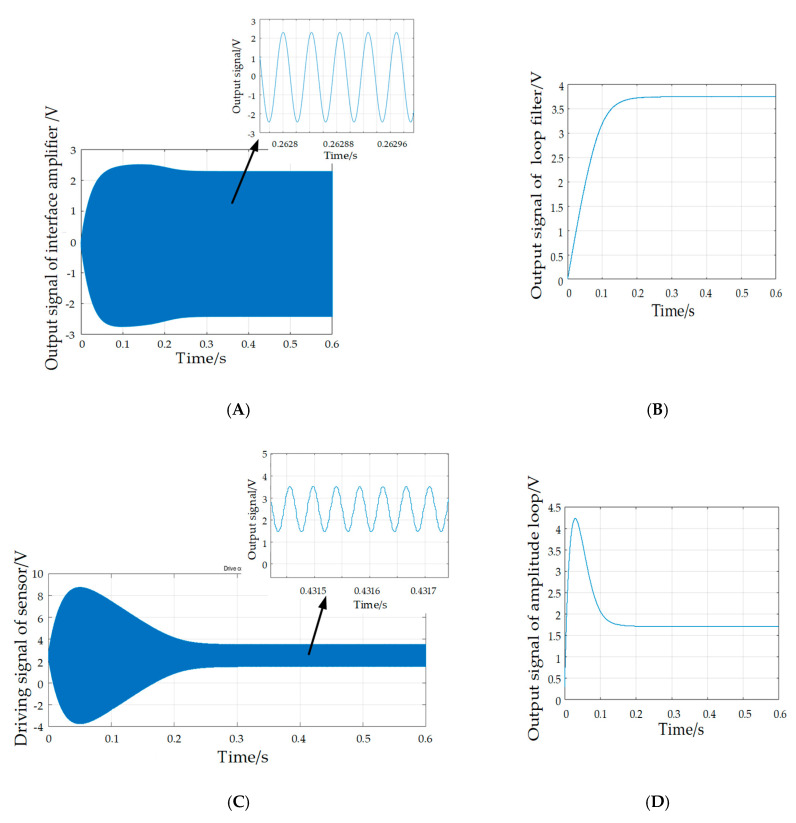
Waveforms of key signals. (**A**) Output of interface amplifier. (**B**) Output signal of loop filter. (**C**) Driving signal of sensor. (**D**) Output signal of amplitude loop.

**Figure 8 sensors-20-04483-f008:**
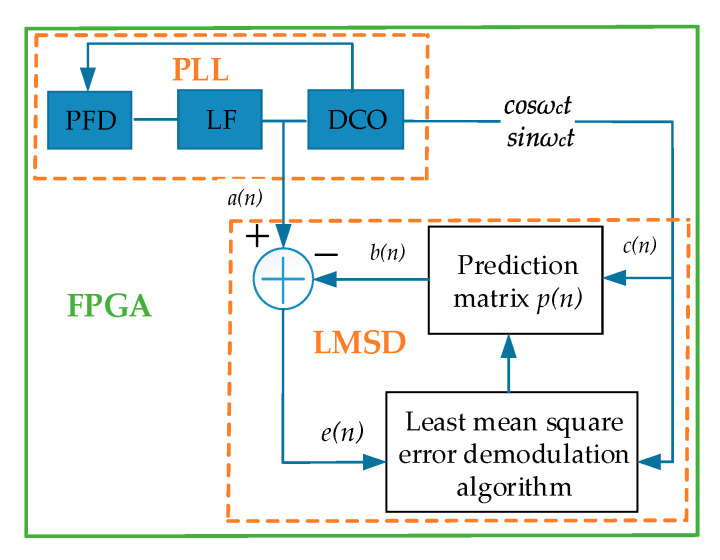
Diagram of least mean square demodulation.

**Figure 9 sensors-20-04483-f009:**
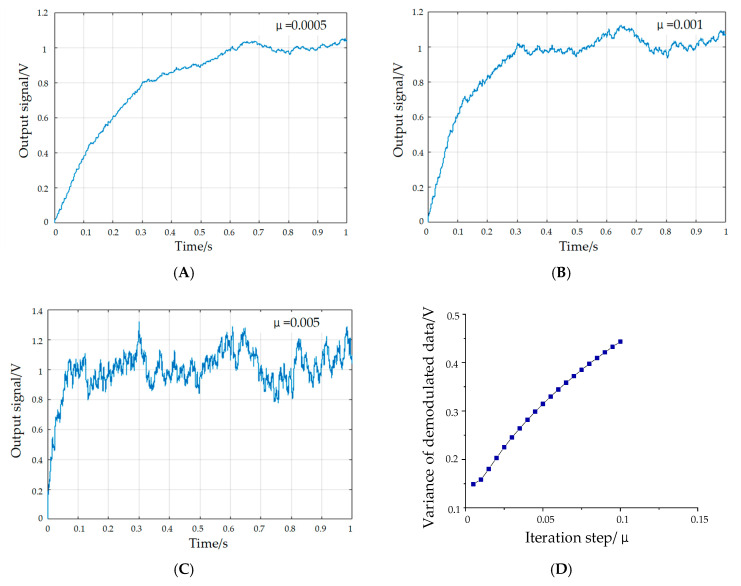
Amplitude curves of demodulation signals. (**A**) *μ* = 0.0005. (**B**) *μ* = 0.001. (**C**) *μ* = 0.005. (**D**) Effect of iteration step on convergence precision.

**Figure 10 sensors-20-04483-f010:**
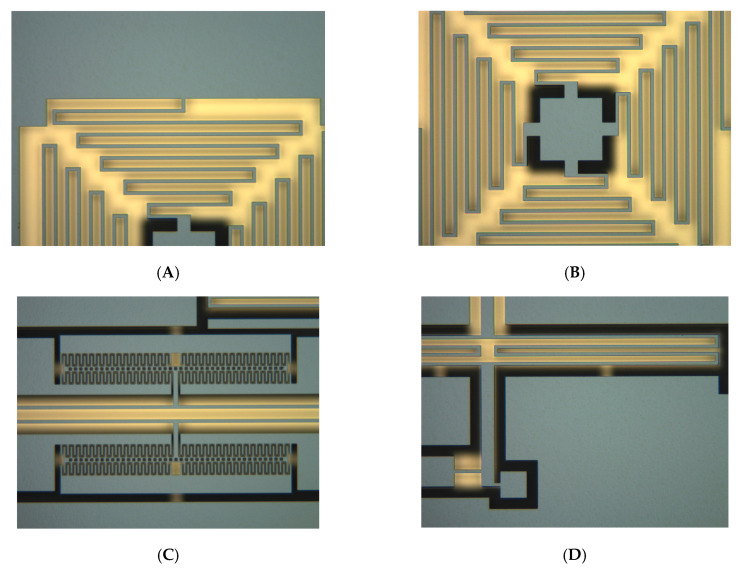
The optical photographs of the fabricated prototype. (**A**) Swing suppression structure. (**B**) The center of the swing suppression structure. (**C**) The double-ended tuning fork resonator. (**D**) The lever elastic beam.

**Figure 11 sensors-20-04483-f011:**
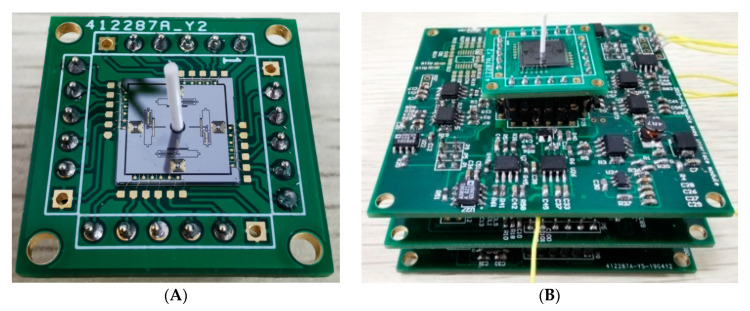
Peripheral hardware. (**A**) Interface board. (**B**) Fabricated control and measurement circuits.

**Figure 12 sensors-20-04483-f012:**
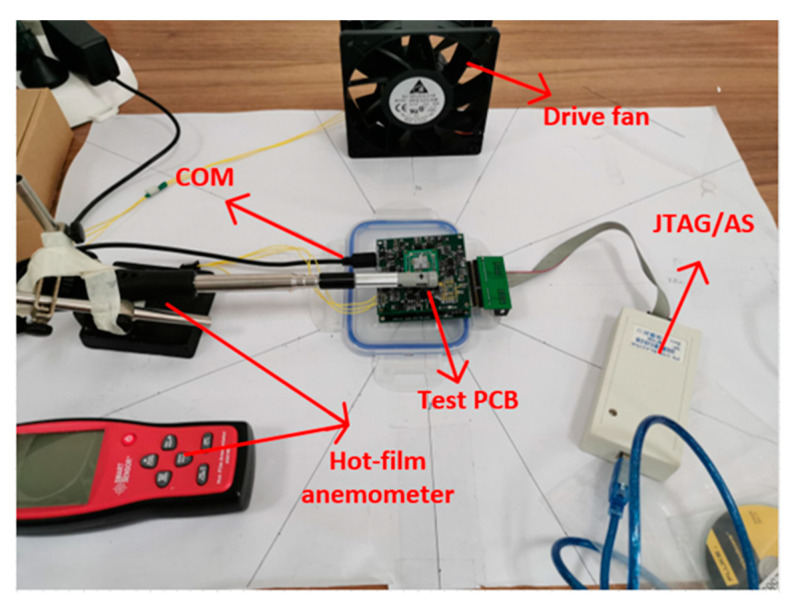
Measurement setup for steady air flow performance evaluation.

**Figure 13 sensors-20-04483-f013:**
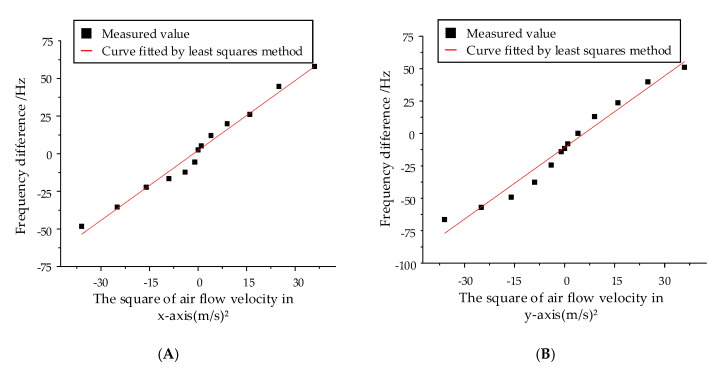
The outputs of the hair flow sensor under steady air flow from 1 m/s to 6 m/s. (**A**) Sensitivity on the x-axis. (**B**) Sensitivity on the y-axis.

**Figure 14 sensors-20-04483-f014:**
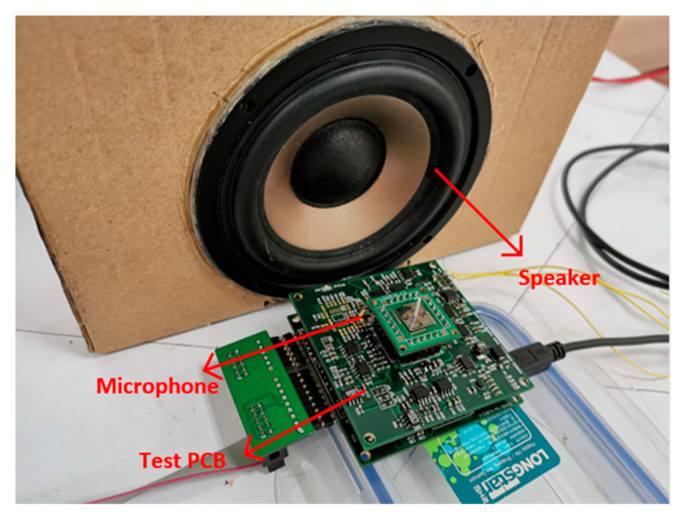
Measurement setup for oscillatory air flow performance evaluation.

**Figure 15 sensors-20-04483-f015:**
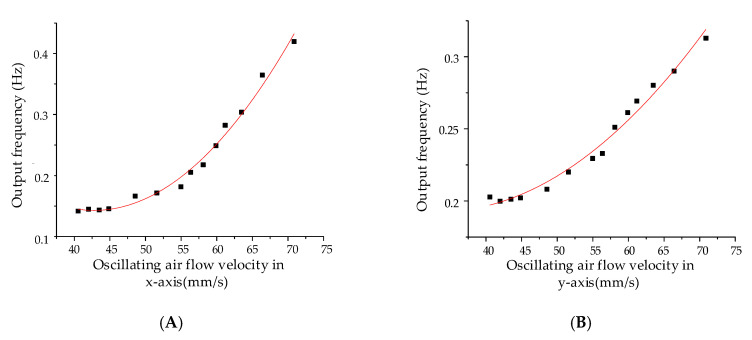
The outputs of the hair flow sensor under oscillatory air flow. (**A**) Sensitivity on the x-axis. (**B**) Sensitivity on the y-axis.

**Table 1 sensors-20-04483-t001:** Structure parameters.

Parameter	Value	Parameter	Value
Resonant beam length (µm)	1300	Hair diameter *D* (µm)	1000
Resonant beam width (µm)	12	Hair height *L_h_* (µm)	9000
Resonant beam thickness (µm)	100	Q-factor of resonator *Q*	43
Lever elastic beam length (µm)	1700	Damping ratio of resonator *ξ*	0.0116
Lever elastic beam width (µm)	12	Proof mass of resonator *m* (kg)	3.32 × 10^−8^
Lever elastic beam thickness (µm)	100	Damping coefficient of resonator *c* (N·s/m)	1.22 × 10^−5^
Connection beam length (µm)	1000	Elasticity coefficient of resonator *k* (N/m)	82.268
Connection beam width (µm)	50	Nature frequency of resonator *w_d_* (rad/s)	25,039 × 2π
Swing suppression beam length (µm)	960	Air density *ρ* (kg/m^3^)	1.293
Swing suppression beam width (µm)	12	Amplification coefficient *A*_1_	26.357
Swing suppression beam thickness (µm)	100	Attenuation coefficient *η*	0.19
Lever support beam length L_1_ (µm)	20	Constant associated with DETFs *t*	3.505
Lever output beam length L_2_ (µm)	100	Constant associated with the device structure *δ*	17.552
Lever output arm length L_3_ (µm)	50	Theoretical mechanical sensitivity *S_ω_* (Hz/(m/s)^2^)	1.583
Lever width L_4_ (µm)	50		

**Table 2 sensors-20-04483-t002:** Effective modes of the novel biaxial bionic hair flow sensor.

Modal	1	2	3	4	5	6
Frequency (Hz)	344.19	344.49	25,106	25,108	25,109	25,113

**Table 3 sensors-20-04483-t003:** Simulation parameters.

Parameter	Value
Cut-off frequency of LF f_c_ (Hz)	75
Sampling time of the system T_s_ (s)	1 × 10^−6^
Scale factor of amplitude closed-loop K_p1_	0.1
Integral factor of amplitude closed-loop K_i1_	2.2 × 10^−4^
Sensitivity of VCO signal (Hz/V)	50
Initial frequency of VCO (Hz)	25,000
Simulation coefficient of drag force F_d_ (N)	5.438 × 10^−9^

**Table 4 sensors-20-04483-t004:** Performance indicators.

Performance Indicators		x-Axis	y-Axis
Mechanical Sensitivity (Hz/(m/s)²)	Theoretical calculated	1.583	1.583
	Measured	1.56	1.81
	Difference	0.02	0.22
Natural frequencies of resonator beams (Hz)	Theory	25,039	
	Simulated	~25,110	
	Measured	23,706, 24,225	23,813, 23,396
Non-linearity (%)	Measured	9.175	7.626
Threshold (mm/s)	Measured	43.27	41.85

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
