# Peer review of "Design and Characterization of a Novel Biaxial Bionic Hair Flow Sensor Based on Resonant Sensing"

_sensors, 2020, doi:10.3390/s20164483_

Round 1

Reviewer 1 Report

A bionic hair flow sensor is described based on a rigid post mounted on a compliant membrane equipped with resonant strain gauges. Although the approach is interesting, it is not really novel, since the authors published a similar design recently. Some further major deficiencies are listed below:

  1. Authors need to refer to their recent publications on the topic, e.g., Bo Yang et al. “Research on an Artificial Lateral Line System Based on a Bionic Hair Sensor with Resonant Readout”, Micromachines 2019, 10, 736; doi:10.3390/mi10110736 and Bo Yang et al.,” A novel flow sensor based on resonant sensing with two-stage microleverage mechanism”, Rev. Sci. Instrum. 89, 045002 (2018); https://doi.org/10.1063/1.5000506. Furthermore, air flow und particle sensing using a piezoresistive strain gauges on a cantilever should be mentioned in the literature survey: Hidetoshi Takahashi et al., “Simultaneous detection of particles and airflow with a MEMS piezoresistive cantilever”, Meas. Sci. Technol. 24 (2013) 025107 (7pp) doi:10.1088/0957-0233/24/2/025107. And piezoresistive cantilevers operated in a resonant mode were also reported to detect airborne nanoparticles, Andi Setiono et al., “In-plane and out-of-plane MEMS piezoresistive cantilever sensors for nanoparticle mass detection”, Sensors 20(3) (2020) 618 (18pp); https://doi.org/10.3390/s20030618.
  2. Basic function of the device and components are not well described, e.g. representation of essential parts in Fig.1 is not enough enlarged.
  3. In their derivation of analytical formulae for describing air flow measurement a textbook reference should be given. What is meant with “the axial force”?
  4. The parameter “air flow rate” is not correctly introduced. Is it a “mass flow rate” or a “volume flow rate”. Furthermore, it is given with the unit (m/s) which, however, is the unit of the “(air) velocity”. Then, on the abscissa of Figs. 5 and 13 a square of air flow rate is given in (m/s^2). This creates much confusion and description has thus to be considerably improved.
  5. In the chapter on mode simulation the “first and second modes of the device” are mentioned, which are obviously not harmonics of a fundamental mode as might be expected. I suppose they represent modes of vibrations in x- and y-direction. But why they do not have identical frequencies, which can be expected, if the structure has an ideal symmetrical shape? Accordingly, the resonance frequencies of the four sensing structure should be identical. What do you mean with “interference mode” and “antiphase mode”?
  6. The representation of the simulation results in Fig. 2 are not meaningful. Fig. 2 can be skipped. Consistence with a theoretical analysis is claimed but not shown.
  7. The definition of sensitivity is not clear and should be given.
  8. The torque value in Table 3 does not have a unit?
  9. In chapter 3.3, the symbols x, y, and z are used to define signals. This may lead to confusion with respect to the labelling of the used coordinate system. This chapter may be shortened, it repeats already published matter (see above).
  10. The description of sensor fabrication at the beginning of chapter 4 is very short without reference to published work.
  11. The double-ended tuning is not completely visible in Fig. 10 (B).
  12. The fabricated sensor has different sensitivities in x- and y-direction. Should be discussed. Limits of uncertainty should be given.
  13. In chapter 4.2 the authors describe a particle vibration/velocity. Which particles are meant here?
  14. An analytical formula is given for the “particle vibration velocity” without derivation and reference to a textbook.
  15. Figure 16 referred to in the text is lacking. An “optimum detecting threshold” is mentioned but not defined. Should the expected characteristic in Fig. 15 show a linear or square dependence?

Reviewer 2 Report

The authors developed a hair like velocity sensor for flow measurement. This concept is not new and the authors are required to do a much thorougher literature survey. Just to quote some former works:

  • Han, Zhiwu, et al. "Artificial hair-like sensors inspired from nature: A review." Journal of Bionic Engineering 15.3 (2018): 409-434.
  • Nawi, Mohd Norzaidi Mat, et al. "Review of MEMS flow sensors based on artificial hair cell sensor." Microsystem technologies 17.9 (2011): 1417.
  • Engel, J. M., et al. "Polyurethane rubber as a MEMS material: characterization and demonstration of an all-polymer two-axis artificial hair cell flow sensor." 18th IEEE International Conference on Micro Electro Mechanical Systems, 2005. MEMS 2005.. IEEE, 2005.
  • Dagamseh, A. M. K., et al. "Engineering of biomimetic hair-flow sensor arrays dedicated to high-resolution flow field measurements." SENSORS, 2010 IEEE. IEEE, 2010.
  • Droogendijk, Harmen, et al. "Advantages of electrostatic spring hardening in biomimetic hair flow sensors." Journal of microelectromechanical systems 24.5 (2015): 1415-1425.

etc.

The structure suggested by the author is not new. There are force sensors with the same structure starting from:

Beccai, Lucia, et al. "Design and fabrication of a hybrid silicon three-axial force sensor for biomechanical applications." Sensors and Actuators A: Physical 120.2 (2005): 370-382.

A good summary of these sensor can be seen in:

Baki, Péter, Gábor Székely, and Gábor Kósa. "Design and characterization of a novel, robust, tri-axial force sensor." Sensors and actuators A: physical 192 (2013): 101-110.   The authors do not explain why their sensor is different from former sensing systems i.e. what is their novelty/contribution.   The theoretical analysis is not good. The approximations of the natural frequency of the beam and the fluid-structure introduction adequate. I would use an Euler Bernoulli beam model to calculate the natural frequency. Figure 2 is not clear, especially the panels C-H. The closed loop sensing is well described. I am missing the data regarding the performance of the circuit such as the Bandwidth of the sensor, theoretical range of the sensor, non-linearity etc. Fig. 13 - please correct the X-axis The experiment is well done. The authors should interpret the results such as the sensor accuracy, non linearity etc. The conclusions do not provide any insight regarding the sensor. For example comparison the insect hair sensors.

Round 2

Reviewer 1 Report

The authors took almost all my comments into account for their revision. However, further amendments will still be necessary:

  1. cf. 4) authors should check, if they change “flow rate” into “flow velocity” everywhere in the paper. E. g., the insets of Figs. 3 a and b still show “flow rate”. By the way, there is an incomplete term of unit (10m/) in these insets.
  2. cf. 6) Figure 2 was meant in my comment: The message of this figure is not clear. It obviously show results of FEM, but details are almost not discernible and it is not clear which details are important. So my question remains: Shouldn’t this figure be skipped?
  3. cf. 7) A definition of sensitivity in an additional formula (not only referring to (6)) as well as values of sensitivity (including uncertainty) of the developed sensor should be given. This is mandatory for a scientific paper in a Sensor Journal. Furthermore, a direct comparison with published data should be given, preferentially in a Table.

Reviewer 2 Report

Regarding nonlinearity - please provide it in % not in absolute value.

Please characterize the results of Fig. 15.: accuracy, fit quality of the curve.

I did not see also any comparison to theoretical calculations.

Please do a thorougher analysis to the results of your sensor.
